# An umbrella review of the evidence associating diet and cancer risk at 11 anatomical sites

Nikos Papadimitriou[1,2], Georgios Markozannes [1,3], Afroditi Kanellopoulou[1], Elena Critselis[4], Sumayah Alhardan[3], Vaia Karafousia[1], John C. Kasimis[1], Chrysavgi Katsaraki[4], Areti Papadopoulou [1], Maria Zografou[1], David S. Lopez[5], Doris S. M. Chan[3], Maria Kyrgiou [6,7], Evangelia Ntzani[1,8], Amanda J. Cross[3,9], Michael T. Marrone[10], Elizabeth A. Platz [10,11,12], Marc J. Gunter[2] & Konstantinos K. Tsilidis [1,3✉]

There is evidence that diet and nutrition are modifiable risk factors for several cancers, but associations may be flawed due to inherent biases. Nutritional epidemiology studies have largely relied on a single assessment of diet using food frequency questionnaires. We conduct an umbrella review of meta-analyses of observational studies to evaluate the strength and validity of the evidence for the association between food/nutrient intake and risk of developing or dying from 11 primary cancers. It is estimated that only few single food/nutrient and cancer associations are supported by strong or highly suggestive meta-analytic evidence, and future similar research is unlikely to change this evidence. Alcohol consumption is positively associated with risk of postmenopausal breast, colorectal, esophageal, head & neck and liver cancer. Consumption of dairy products, milk, calcium and wholegrains are inversely associated with colorectal cancer risk. Coffee consumption is inversely associated with risk of liver cancer and skin basal cell carcinoma.

[1] Department of Hygiene and Epidemiology, University of Ioannina School of Medicine, Ioannina, Greece. [2] Section of Nutrition and Metabolism, International Agency for Research on Cancer, Lyon, France. [3] Department of Epidemiology and Biostatistics, School of Public Health, Imperial College London, London, UK. [4] Proteomics Facility, Center for Systems Biology, Biomedical Research Foundation of the Academy of Athens, Athens, Greece. [5] Department of Preventive Medicine and Population Health, University of Texas Medical Branch, Galveston, TX, USA. [6] Department of Gut, Metabolism and Reproduction and Department of Surgery and Cancer, Institute of Reproductive and Developmental Biology, Faculty of Medicine, Imperial College London, London, UK. [7] West London Gynaecological Cancer Centre, Imperial College Healthcare NHS Trust, London, UK. [8] Center for Evidence-Based Medicine, Department of Health Services, Policy and Practice, School of Public Health, Brown University, Providence, RI, USA. [9] Cancer Screening and Prevention Research Group (CSPRG), Department of Surgery and Cancer, Imperial College London, London, UK. [10] Department of Epidemiology, Johns Hopkins Bloomberg School of Public Health, Baltimore, MD, USA. [11] Sidney Kimmel Comprehensive Cancer Center at Johns Hopkins, Baltimore, MD, USA. [12] Department of Urology and the James Buchanan Brady Urological Institute, Johns Hopkins University School of Medicine, Baltimore, MD, USA. ✉email: ktsilidi@uoi.gr

Cancer is one of the most frequent causes of morbidity and mortality worldwide, and 18.1 million incident cases and 9.6 million deaths were estimated for 2018[1]. The World Cancer Research Fund (WCRF) Third Expert Report concluded that diet and nutrition, including obesity and low physical activity, are modifiable risk factors for several cancers[2]. However, reported meta-analytic estimates from observational studies might not represent causal associations but may emerge due to biases common across studies, such as exposure measurement error, residual confounding, and publication bias, subsequently diminishing the strength of aggregated scientific evidence[3–5].

Nutritional epidemiology is particularly prone to measurement error, as the consumption of foods and nutrients is based on participants' self-reported data often provided at single point in time and the conversion of foods consumed into nutrient intake is based on food composition databases that might be inaccurate[6]. An evaluation of the robustness of this evidence is required to inform public health policy. Therefore, we conducted an umbrella review to systematically evaluate the robustness of the observational meta-analytic evidence across a large number of food and nutrient associations with risk of cancer at 11 anatomical sites. We further evaluated whether additional research is or is not warranted to change the inferences from the existing meta-analyses using an adaptation of research synthesis methods.

Here we show that only few single food/nutrient and cancer associations are supported by strong or highly suggestive meta-analytic evidence, and future similar research is unlikely to change this evidence.

## Results

**Characteristics of the meta-analyses.** Overall, 860 meta-analytic comparisons were included. Most meta-analyses ($n = 779$; 91%) used a continuous exposure contrast, whereas only 81 used a dichotomous contrast (top versus bottom category or use versus no use) (Supplementary Data 1 and 2). The summary descriptive characteristics of the included meta-analyses by cancer type are presented in Table 1. Associations for colorectal cancer (CRC) and breast cancer had the largest number of meta-analyses ($N = 221$ and 163, respectively) followed by lung and stomach cancers ($N = 144$ and 122, respectively). The median number of cancer cases in a meta-analysis ranged from 388 for gallbladder cancer to 4526 for breast cancer. The median number of individual studies within a meta-analysis ranged from 2.5 for gallbladder cancer to 6 for CRC and breast cancer, and the overall minimum number of individual studies was 2 and the maximum was 33. The summary descriptive characteristics of the included meta-analyses by type of dietary exposure are presented in Supplementary Data 3. Associations for fruits/vegetables and alcohol had the largest number of meta-analyses ($N = 184$ and 141, respectively) followed by meat/eggs and beverages ($N = 118$ and 70, respectively). The median number of individual studies within a meta-analysis was 5 and was similar for different dietary exposures with the exception of fats/fatty acids, where the median was 11. The overall median number of cancer cases in a meta-analysis was 2152 and ranged from 1038 for phytochemicals to 9955 for fiber.

**Summary effects and heterogeneity between studies.** Of the 860 meta-analyses, 247 (29%) were nominally statistically significant ($P < 0.05$) based on the random-effects model. The percentage of nominally significant meta-analyses ranged from 0% for gallbladder cancer (which had only two available meta-analyses) to 74% for head and neck cancer (Table 2) and from 0% for legumes/soy products to 54% for alcohol and 67% for grains (which had only three available meta-analyses) (Supplementary Data 4). At the stricter $P$ value threshold of $10^{-3}$, only 75 (9%)

**Table 1 Descriptive statistics of the meta-analyses overall and by cancer type included in the umbrella review grading the evidence on diet and cancer risk.**

| | Total | Head and neck[a] | Esophageal | Stomach | Colorectal | Liver | Gallbladder | Lung | Skin | Breast | Kidney | Urinary bladder |
|---|---|---|---|---|---|---|---|---|---|---|---|---|
| Number of meta-analyses | 860 | 38 | 48 | 122 | 221 | 20 | 2 | 144 | 18 | 163 | 41 | 43 |
| Number of studies | | | | | | | | | | | | |
| Median | 5 | 4 | 3 | 4 | 6 | 4 | 2.5 | 5 | 4 | 6 | 4 | 4 |
| Min-max | 2–33 | 2–23 | 2–17 | 2–23 | 2–25 | 2–14 | 2–3 | 2–32 | 2–7 | 2–33 | 2–16 | 2–11 |
| Number of participants | | | | | | | | | | | | |
| Median | 434,534 | 517,979 | 947,646 | 620,638 | 512,243 | 970,317 | 370,938 | 247,214 | 114,222 | 301,936 | 523,757 | 386,216 |
| Min-max | 236–5,831,157 | 168,996–5,831,157 | 30,776–3,770,260 | 2629–3,948,365 | 1069–5,218,250 | 117,042–4,341,949 | 222,216–519,660 | 559–5,084,099 | 5498–1,741,417 | 236–5,197,851 | 119,537–2,192,726 | 1233–1,630,981 |
| Number of cases | | | | | | | | | | | | |
| Median | 2152 | 700 | 636 | 1301 | 3374 | 848 | 388 | 1935 | 2146 | 4526 | 1437 | 1905 |
| Min-max | 56–117,514 | 56–14,730 | 100–5666 | 125–13,549 | 239–36,942 | 211–5977 | 375–400 | 211–23,607 | 242–33,548 | 81–117,514 | 272–4171 | 265–5409 |

[a]This report presents results for cancers of mouth, pharynx, larynx, and upper aerodigestive tract.

**Table 2 Number and percentage of meta-analyses overall and by cancer type that meet the individual and the overall criteria used for the grading of the evidence on diet and cancer risk.**

| Criterion | Total (n = 860) | Head and neck[a] (n = 38) | Esophageal (n = 48) | Stomach (n = 122) | Colorectal (n = 221) | Liver (n = 20) | Gallbladder (n = 2) | Lung (n = 144) | Skin (n = 18) | Breast (n = 163) | Kidney (n = 41) | Urinary bladder (n = 43) |
|---|---|---|---|---|---|---|---|---|---|---|---|---|
| $P$ value <10$^{-6}$, n (%) | 25 (2.9) | 5 (13.2) | 2 (4.2) | 0 (0) | 12 (5.4) | 2 (10) | 0 (0) | 0 (0) | 1 (5.6) | 3 (1.8) | 0 (0) | 0 (0) |
| $P$ value <10$^{-3}$, n (%) | 75 (8.7) | 11 (28.9) | 6 (12.5) | 2 (1.6) | 24 (10.9) | 6 (30) | 0 (0) | 11 (7.6) | 4 (22.2) | 11 (6.7) | 0 (0) | 0 (0) |
| $P$ value <0.05, n (%) | 247 (28.7) | 28 (73.7) | 17 (35.4) | 19 (15.6) | 69 (31.2) | 10 (50) | 0 (0) | 54 (37.5) | 6 (33.3) | 35 (21.5) | 7 (17.1) | 2 (4.7) |
| $I^2$ >50%, n (%) | 227 (26.4) | 15 (39.5) | 14 (29.2) | 25 (20.5) | 44 (20) | 7 (35) | 0 (0) | 45 (31.3) | 6 (33.3) | 46 (28.2) | 12 (29.3) | 13 (30.2) |
| $I^2$ ≤25%, n (%) | 450 (52.3) | 18 (47.4) | 27 (56.3) | 77 (63.1) | 115 (52.0) | 8 (40) | 2 (100) | 73 (50.7) | 8 (44.4) | 79 (48.5) | 23 (56.1) | 20 (46.5) |
| Prediction interval excluding the null, n (%) | 46 (5.3) | 1 (7.9) | 1 (2.1) | 4 (3.3) | 22 (10) | 2 (10) | 0 (0) | 5 (3.5) | 0 (0) | 10 (6.1) | 0 (0) | 1 (2.3) |
| Evidence of small study bias[b], n (%) | 69 (8.0) | 5 (13.2) | 7 (14.9) | 7 (5.4) | 13 (5.9) | 2 (10) | 0 (0) | 19 (13.2) | 2 (1.1) | 6 (3.7) | 5 (12.2) | 3 (7.0) |
| Evidence of excess significance bias[c], n (%) | 121 (14.1) | 15 (39.5) | 6 (12.5) | 12 (9.8) | 23 (10.4) | 6 (30) | 0 (0) | 32 (22.2) | 4 (22.2) | 20 (12.3) | 3 (7.3) | 0 (0) |
| **Overall grading** | | | | | | | | | | | | |
| Not significant, n (%) | 613 (71.3) | 10 (26.3) | 31 (64.6) | 103 (84.4) | 152 (68.8) | 10 (50.0) | 2 (100) | 90 (62.5) | 12 (66.7) | 128 (78.5) | 34 (82.9) | 41 (95.4) |
| Weak, n (%) | 182 (21.2) | 23 (60.5) | 12 (25.0) | 17 (13.9) | 46 (20.8) | 5 (25.0) | 0 (0) | 44 (30.6) | 2 (11.1) | 24 (14.7) | 7 (17.1) | 2 (4.7) |
| Suggestive, n (%) | 42 (4.9) | 1 (2.6) | 4 (8.3) | 2 (1.6) | 11 (5.0) | 3 (15.0) | 0 (0) | 10 (6.9) | 3 (16.7) | 8 (4.9) | 0 (0) | 0 (0) |
| Highly suggestive, n (%) | 13 (1.5) | 4 (10.5) | 1 (2.1) | 0 (0) | 4 (1.8) | 2 (10.0) | 0 (0) | 0 (0) | 1 (5.6) | 1 (0.6) | 0 (0) | 0 (0) |
| Strong, n (%) | 10 (1.2) | 0 (0) | 0 (0) | 0 (0) | 8 (3.6) | 0 (0) | 0 (0) | 0 (0) | 0 (0) | 2 (1.2) | 0 (0) | 0 (0) |

[a]This report presents results for cancers of mouth, pharynx, larynx, and upper aerodigestive tract.
[b]Small study bias is based on the $P$ value from the Egger's regression asymmetry test (two-sided $P$ value ≤0.1) and the random-effects summary estimate was larger compared to the point estimate of the largest study in a meta-analysis.
[c]Excess significance bias is based on the $P$ value (two-sided $P$ value ≤0.1) of the excess significance test using the largest study (smallest standard error) in a meta-analysis as the plausible effect size.

meta-analyses showed significant results, whereas only 25 (3%) meta-analyses remained significant at a threshold of 10$^{-6}$ (Table 2).

Approximately one fourth (26%) of the included meta-analyses had high heterogeneity ($I^2$ > 50%). Head and neck cancer had the largest proportion of meta-analyses with high heterogeneity (40%) that was substantially reduced after excluding a pooling project of case–control studies, whereas gallbladder cancer had no meta-analysis with high heterogeneity followed by CRC and stomach cancer with 20% each (Table 2). Associations for energy/sugars and vitamin D had the largest proportion of meta-analyses with high heterogeneity (50% each) followed by alcohol (43%) (Supplementary Data 4). The proportion of meta-analyses with little evidence of heterogeneity ($I^2$ ≤ 25%) was >52%. However, only 46 (5%) associations had a 95% prediction interval that excluded the null value (Table 2).

**Small study effects and excess significance bias.** Sixty-nine (8%) meta-analyses showed evidence of small study effects bias (Table 2), which was highest for salt/salty foods (25%; only 4 available meta-analyses) followed by alcohol (16%) and fruits/vegetables (13%) (Supplementary Data 4). Additionally, the proportion of meta-analyses showing evidence of excess significance bias was 14% and ranged from 0% for urinary bladder and gallbladder cancers to 40% for head and neck cancer (Table 2) and was highest for associations that involved alcohol (35%) and salt/salty foods (50%) (Supplementary Data 4).

**Grading of the evidence.** Only 10 meta-analyses (1.2%) were supported by strong evidence (Figs. 1 and 2 and Supplementary Data 1 and 2), and represented associations for the following dietary intakes and risk of CRC and breast cancer: alcohol ($n = 5$ associations), dietary calcium ($n = 1$), dairy products ($n = 2$), milk ($n = 1$), and whole grain products ($n = 1$). Specifically, total alcohol consumption was positively associated with risk of CRC (summary relative risk (RR) per 10 g/day [~1 drink], 1.07; 95% confidence interval (CI), 1.05–1.08), and identical associations were observed for beer consumption and CRC and total alcohol consumption with colon cancer risk. Consumption of dairy products (RR per 400 g/day [~2 servings], 0.87; 95% CI, 0.83–0.90), milk (RR per 200 g/day, 0.94; 95% CI, 0.92–0.96), calcium (RR high versus low, 0.83; 95% CI, 0.79–0.87), and whole grains (RR per 90 g/day [~3 servings], 0.84; 95% CI, 0.78–0.90) was inversely associated with risk of CRC. Total alcohol consumption (RR per 10 g/day, 1.12; 95% CI, 1.09–1.15) and similarly wine consumption were positively associated with risk of postmenopausal breast cancer among current users of menopausal hormone therapy and postmenopausal breast cancer, respectively.

Thirteen meta-analyses (1.5%) presented highly suggestive evidence (Figs. 1 and 2 and Supplementary Data 1 and 2), and most pertained to alcohol ($n = 9$), followed by coffee ($n = 2$), fruits ($n = 1$), and vegetables ($n = 1$). Alcohol consumption was positively associated with different subtypes of breast cancer and CRC, esophageal cancer in men (RR per 10 g/day, 1.33; 95% CI, 1.22–1.46), head and neck cancer (RR for oral cancer, 1.15; 95% CI, 1.09–1.22; RR for upper aerodigestive tract cancer, 1.18; 95% CI, 1.11–1.26), and liver cancer mortality (RR, 1.02; 95% CI, 1.01–1.03). Coffee consumption was inversely associated with risk of liver cancer (RR per 1 cup/day, 0.85; 95% CI, 0.81–0.90) and skin basal cell carcinoma (RR, 0.95; 95% CI, 0.94–0.97), and intake of fruits and vegetables was inversely associated with risk of pharyngeal (RR high versus low, 0.60; 95% CI, 0.52–0.70) and oral cancer (RR high versus low, 0.68; 95% CI, 0.60–0.77), respectively. When we excluded a pooling project of 20

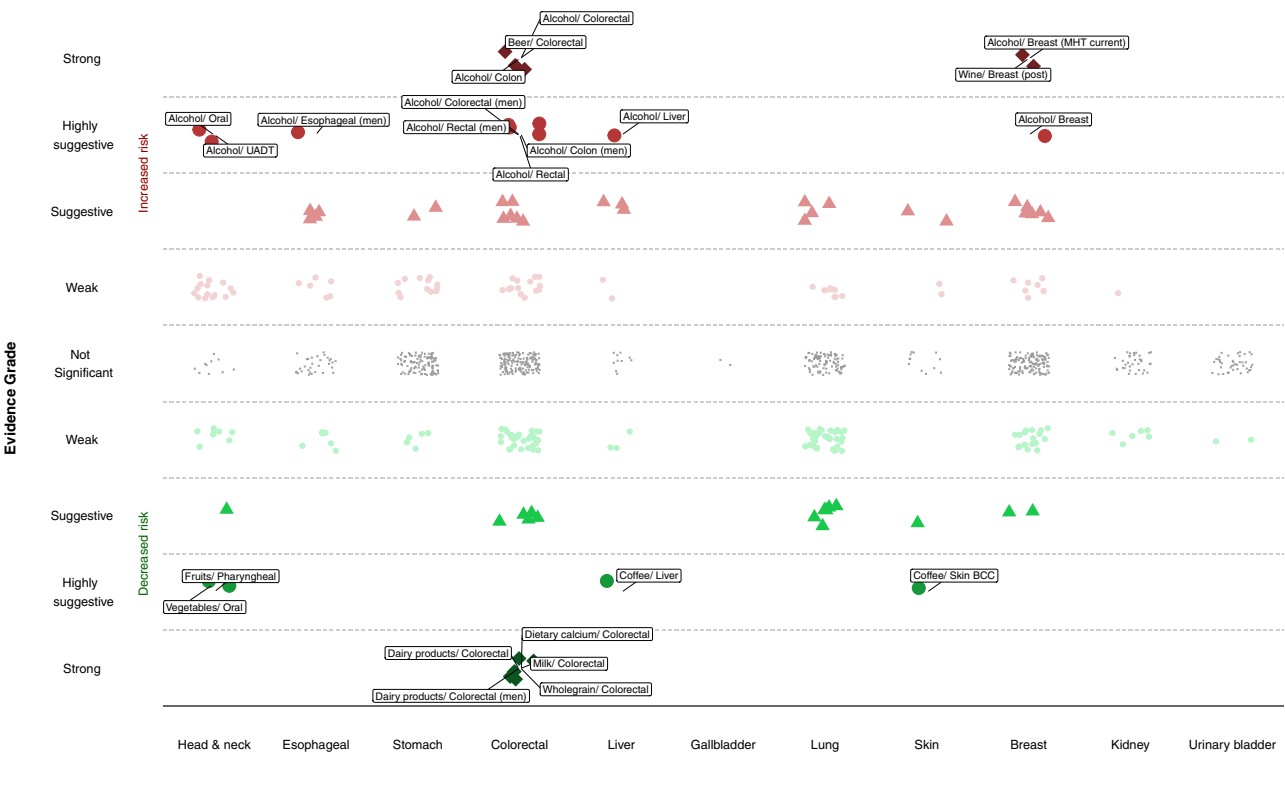

**Fig. 1 Scatter plot showing results from the umbrella review grading the evidence on diet and cancer risk.** The *Y*-axis shows the strength of the evidence. The upper half displays associations that increase cancer risk (in red), whereas the bottom half shows associations that reduce cancer risk (in green). The different point symbols and color intensity represent the different levels of evidence grading. Points colored in gray denote no statistically significant associations. The *X*-axis corresponds to the different cancer sites. BCC basal cell carcinoma, MHT menopausal hormone therapy, UADT upper aerodigestive tract.

case–control studies from the meta-analyses of dietary factors with risk of head and neck cancer, we observed an attenuation of the evidence for most of the results (Supplementary Data 1). The protective association of fruit consumption on pharyngeal cancer was no longer statistically significant, and the inverse association of vegetables and oral cancer was downgraded to weak evidence. Among the associations supported by strong or highly suggestive evidence, the majority of individual studies in each respective meta-analysis had adjusted for smoking (67–100%) and adiposity (67–95%), but adjustment for physical activity was less frequently conducted (5–79%) (Supplementary Data 2).

Forty-two (4.9%) meta-analyses had suggestive evidence, 18 of which pertained to associations that already received strong or highly suggestive evidence but were performed for different types of the same exposure and/or different subtypes of same outcome (Fig. 1 and Supplementary Data 1 and 2). Notable additional associations that received suggestive evidence were: (i) positive associations of red and/or processed meat consumption and risk of CRC (*n* = 5 associations; RR for processed meat per 50 g/day and CRC, 1.16; 95% CI, 1.08–1.26; RR for red/processed meat per 100 g/day and colon cancer, 1.19; 95% CI, 1.10–1.29), (ii) inverse associations of total dietary (RR per 10 g/day, 0.95; 95% CI, 0.93–0.98) and soluble fiber (RR, 0.75; 95% CI, 0.63–0.88) and risk of breast cancer, (iii) inverse associations of serum retinol and α-carotene concentrations and fruit, folate, and vitamin C consumption with risk of lung cancer, (iv) positive associations of alcohol and red/processed meat with lung cancer, (v) inverse association of coffee consumption and melanoma risk in women, (vi) positive associations of alcohol consumption and serum vitamin D with basal cell carcinoma of skin, and (vii) positive

associations for pickled vegetables and salty foods and risk of stomach cancer. A total of 182 meta-analyses (21%) were supported by weak evidence, and the remaining 613 (71%) meta-analyses did not have nominally statistically significant findings.

**Number of additional studies needed to change current meta-analytic evidence**. Among the 65 meta-analyses that achieved strong, highly suggestive, or suggestive evidence, the median fail-safe number (FSN) was 23 (range: 4–159) for meta-analyses with suggestive evidence, 111 (range: 38–856) for highly suggestive and 67 (range: 32–369) for strong evidence (Supplementary Data 1). The FSN was always larger than the number of studies included in the current meta-analyses for these evidence categories, suggesting that the addition of only a likely implausible number of further studies could drive the statistically significant summary estimates to null. For the 182 meta-analyses that were supported by weak evidence, the median FSN was 4 (range: 1–42), and the FSN was smaller than the number of studies included in the current meta-analyses for 105 comparisons, further supporting their weak evidence for association.

The majority of the 613 non-statistically significant meta-analytic comparisons required >10 additional studies to achieve at least 80% conditional power (CP) to detect a statistically significant summary estimate using either the random-effects estimate (479; 78%) or the estimate of the largest study (473; 77%) (Supplementary Data 1), suggesting potential futility of additional research. Only in 28 (5%) associations, the estimated number of additional studies that could provide sufficient power to drive

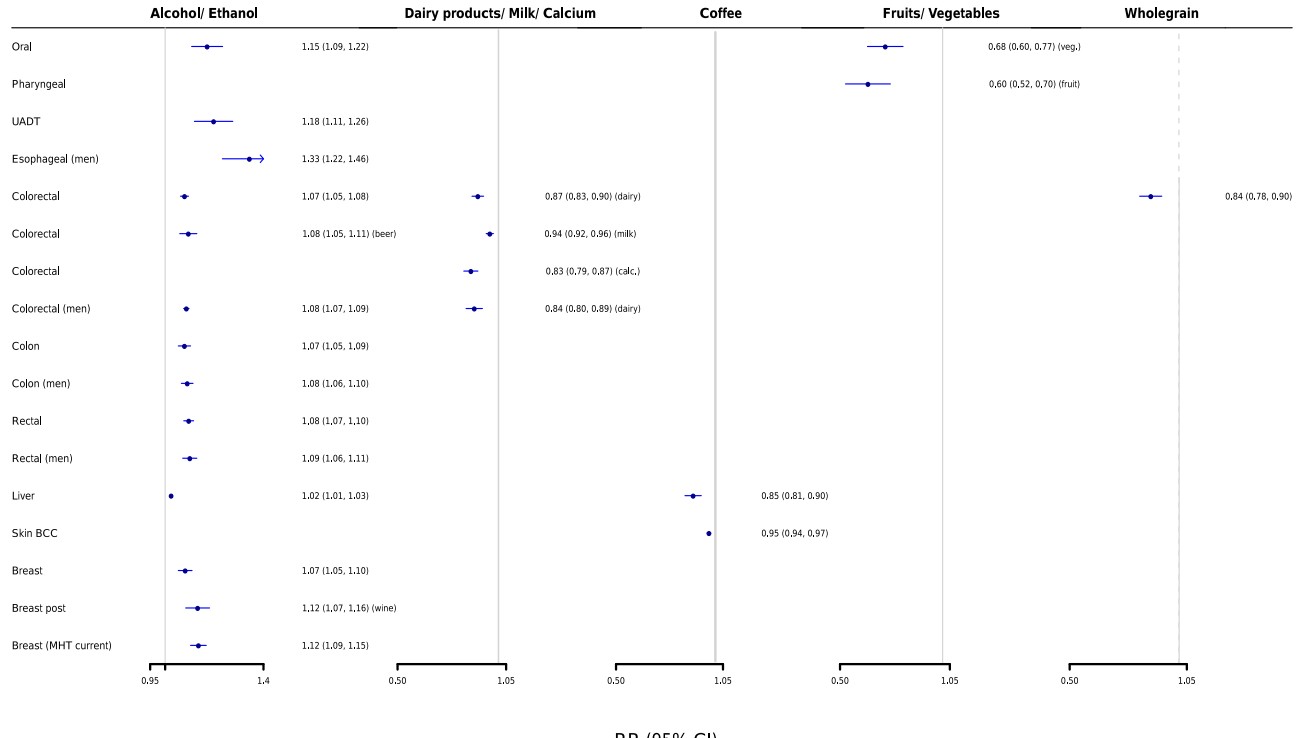

**Fig. 2 Forest plot showing results that achieved strong or highly suggestive evidence from the umbrella review on diet and cancer risk.** Data are presented as relative risks and 95% confidence intervals. BCC basal cell carcinoma, MHT menopausal hormone therapy, UADT upper aerodigestive tract.

summary estimates to nominal significance was less than the number of studies included in the current meta-analyses, suggesting further study could change inferences (Supplementary Data 5). These associations pertained mostly to understudied cancers, namely, esophageal (citrus fruits), stomach, (processed meat, egg, non-fermented soya, coffee), rectal (red/processed meat, poultry, fiber, serum folate), kidney (red/processed meat, fruits, vegetables, tomatoes, coffee), and urinary bladder (vegetables), and fewer associations were reported for colon (folate [CRC], glycemic load, tea), lung (vegetables, egg), and breast (fat, coffee, vegetables, folate, plasma α-carotene) cancer.

## Discussion

**Main findings**. We reviewed 860 meta-analyses from the WCRF Third Expert Report to evaluate the quality of evidence for associations between a large range of dietary factors and risk of cancer at 11 anatomical sites. Ten associations were supported by strong meta-analytic evidence inferred by strongly statistically significant results and no suggestion of bias. These associations were between alcohol consumption and higher risk of CRC and breast cancer, as well as between calcium, dairy, and whole grain products and lower risk of CRC. Thirteen associations were supported by highly suggestive evidence, and most pertained to alcohol and higher risk of cancers of the colon, rectum, esophagus, head and neck, and liver. Additionally, consumption of coffee was inversely associated with risk of liver and skin basal cell carcinoma, and intake of fruits and vegetables were inversely associated with head and neck cancer risk.

Several dietary factors have been clearly associated with risk of obesity, type 2 diabetes, and cardiovascular disease[7]. Although nutrition-related factors, such as obesity and lack of physical activity, are established risk factors for several cancers[2,8,9], the association of specific dietary factors with cancer risk is less well recognized and potentially biased due to exposure measurement

error and reporting biases[10,11]. The extent to which this literature is affected by biases is difficult to prove. We used statistical tests and sensitivity analyses to look for evidence of bias. Overall, a large number of meta-analyses ($N = 860$) was evaluated, but they contained on average relatively few studies (median = 5) and a moderate number of cancer cases (median = 2152). Less than 30% of the included associations on diet and cancer risk reported a nominally statistically significant summary random-effects estimate. When a lower $P$ value threshold ($P < 10^{-6}$) was used, the proportion of significant associations decreased to 3%, indicating potential dearth of existing robust associations. Approximately 1 in 4 associations showed large heterogeneity ($I^2 \geq 50\%$). When we calculated the 95% prediction intervals, which further account for heterogeneity, we found that the null value was excluded in only 5% of the associations, an estimate that was mostly driven by the low percentage of nominally significant meta-analyses. Moreover, in 8% of the associations the summary estimates were inflated due to small study effects and in 15% the observed number of significant results was larger than the expected, indicating that the risk of reporting and other biases in this literature is relatively low, yet these tests had low power, as many meta-analyses included a small number of studies. The latter summary description of the robustness of the evidence seems to apply for most dietary exposure categories and cancer outcomes, with one notable exception. A larger proportion (11%) of alcohol and cancer associations (compared to other diet-cancer associations) is supported by strong statistical evidence, but potential evidence for existence of bias is also present in the alcohol and cancer associations shown by larger likelihoods of high heterogeneity, small study effects, and excess significance bias. When the FSN and CP research synthesis metrics were used, they indicated that additional similar research is unlikely to change current evidence for most associations with few exceptions that pertained mostly to currently observed null associations between dietary factors and understudied malignancies.

**Comparison with other evidence gradings**. Our grading of the evidence suggested that there are a limited number of dietary factors that are robustly associated with specific cancers, which agrees with the evidence grading performed independently by the WCRF[2]. Specifically, most associations that were supported by strong or highly suggestive evidence in the current umbrella review were also supported with convincing evidence by WCRF. Exceptions were the inverse associations of coffee consumption with skin cancer and fruit and vegetable consumption with head and neck cancer, which received limited suggestive evidence by WCRF and highly suggestive evidence in this umbrella review, but the majority of the evidence for associations with head and neck cancer stemmed from case–control studies. There were a few associations that received convincing evidence grading by WCRF but were ranked lower in the current umbrella review. Specifically, we graded the consumption of red and/or processed meat with risk of CRC and salty foods with stomach cancer risk with suggestive evidence, because they had summary association $P$ values of approximately $10^{-4}$ and did not reach the strict threshold of $10^{-6}$. The recent evaluation of the literature for the health effects of red and processed meat consumption conducted by the NutriRECS consortium yielded very similar (to this umbrella review) positive associations for red/processed meat and CRC risk[12] but graded the certainty of evidence as low, because they used the GRADE system to assess quality of evidence where evidence from observational studies is ranked lower compared to randomized controlled trials (RCTs)[13]. The associations of alcohol intake with risk of kidney and stomach cancer were graded with weak evidence in the current umbrella review, because they had summary $P$ values of $10^{-2}$ and $10^{-3}$ but also high between-study heterogeneity and evidence of excess significance bias. The association of alcohol consumption with risk of premenopausal breast cancer was graded with convincing evidence by WCRF, but this assessment did not consider a more recent pooled analysis that rendered the overall association as not significant[14].

It has been estimated that diet and nutrition could account for 20–25% of the worldwide cancer burden[15,16]. The obesogenic effects of a high calorie diet and lack of physical activity could account for about 10–15% of the cancer burden, whereas about 5% may be attributable to alcohol and another 5% to specific dietary factors combined (e.g., red meat, whole grains, calcium)[15,16]. The current umbrella review supports the notion that there are a limited number of energy-balance independent dietary factors that are robustly associated with cancer risk. However, it is critical to continue and enhance research efforts and investments in this field, because diet is a ubiquitous exposure and changes in diet may modify cancer risk. The conduct of nutritional epidemiologic research on cancer has many limitations and challenges. Cancer is a group of diseases with long latency periods, but assessment of dietary intakes is usually performed only at one or at few time points in several well-established cohort studies, which may lead to underestimation of some diet and cancer associations. In addition, previous research has mostly focused on studying associations of single nutrients and foods, for which it may be difficult to decipher their independent effects and may not be biologically important in isolation, but synergistic effects of numerous foods into overall dietary patterns may matter more.

**Current evidence and mechanisms**. There is ample evidence in agreement with the results of the current umbrella review suggesting that alcohol is a major risk factor for several cancers, including breast, CRC, esophageal, head and neck, and liver cancer[2,17]. Biological mechanisms linking alcohol consumption to breast cancer would mainly involve altered circulating and intracellular estrogen concentrations and subsequent proliferation of estrogen receptors (ER) in mammary epithelial cells[18]. More specifically, alcohol consumption has been associated with increased circulating estrogen and androgen concentrations in observational studies and RCTs, and ethanol promotes the proliferation of ER$^+$ but not ER$^-$ breast cancer cells leading to a 10–15-fold increase in the transcriptional activity of ER[18]. Chronic alcohol intake has been further associated with oxidative stress, intestinal dysbiosis, and hyperpermeability to luminal bacterial products, which may lead to the development of CRC[19–21], alcoholic liver disease, and liver cancer. The direct carcinogenic effects of acetaldehyde and its metabolites is another potential mechanism for cancer onset, as acetaldehyde rapidly binds to DNA and proteins and produces DNA adducts, which results in DNA point mutations[18].

The current literature supports an inverse association between consumption of dairy foods and CRC risk[22], in agreement with the results of the current umbrella review. Dairy products may lead to lower CRC risk due to their high calcium concentration[23]; calcium forms in the colonic lumen insoluble soaps by binding to tumor-promoting free fatty acids and bile acids[21,24]. Other pathways include regulation of cell proliferation, differentiation and apoptosis through the preservation of intestinal epithelial cell integrity, and maintenance of immune homeostasis in the gut[21]. Apart from the role of calcium, studies have shown that lactic acid bacteria may absorb mutagens from cooked foods, deactivate intestinal carcinogens, and reduce intestinal inflammation[21,25].

The current umbrella review found strong evidence to support the inverse association of whole grain products and CRC risk. Whole grain products are rich in dietary fiber and other nutrients and substances with potential anticancer properties[26,27]. Studies have shown that consumption of whole grains reduce fasting insulin concentrations, which is an established risk factor for CRC[21]. In addition, dietary fiber shortens the bowel transit time, dilutes the colonic contents, and promotes the anaerobic fermentation of the intestinal bacteria[28,29]. As a result, the carcinogenic substances are in contact with epithelial cells for a short period of time, while short chain fatty acids like butyrate are produced that reduce the conversion of primary bile acids to secondary and also inhibit cell proliferation and promote cell apoptosis[28,29].

Coffee is one of the most commonly consumed beverages at a global level and its association with several outcomes has been extensively investigated[30]. We found highly suggestive evidence that coffee intake was inversely associated with risk of liver and skin basal cell carcinoma. The beneficial effects of coffee consumption might be due to the antioxidant and anti-inflammatory properties of its phytochemical compounds that may protect against diseases triggered by inflammation like cancer[30]. Additionally, coffee consumption has been linked with a better profile of markers of liver injury that could be another mechanism though which coffee may lower liver cancer risk[31,32].

**Strengths and weaknesses**. We applied several statistical criteria and sensitivity analyses to evaluate the strength and validity of the evidence for the association between food and nutrient intake and risk of 11 cancers using the most extensive systematic review to date from the WCRF Continuous Update Project (CUP). Several different methods exist for rating quality of evidence[33], but they are inconsistent and allow some degree of arbitrariness. The criteria we used for grading evidence should not be considered causal criteria, especially when used individually, but we think that they are useful for identifying biases when used together.

We further applied two established research synthesis metrics, the CP and FSN, to determine whether or not further research can provide a meaningful contribution to the existing meta-analytic evidence in an effort to guide the future research agenda and public health policy. The application of these metrics should not be interpreted as stopping research altogether but rather to focus future research to address current evidence gaps.

Important limitations should also be considered in the interpretation of our findings. Our review relied on already published studies that were included in the meta-analyses performed by the WCRF CUP through 2018. Some studies may have been missed, although this is unlikely given the extensive literature search conducted. We evaluated all study-specific results that were reported in the meta-analyses, namely, primary cancer incidence and/or mortality, histological and anatomical cancer subtypes, and analyses by sex, menopausal status, smoking, and hormone replacement therapy, but we may have missed other sub-analyses that were not reported with sufficient study-specific data.

In addition, the current umbrella review was based on results derived only from observational studies. Evidence from RCTs is essential to identify potential causal associations in epidemiology. However, there is a lack of adequately powered RCTs in the field of nutritional epidemiology, and those that do exist have in general failed to support protective associations[34–37]. Furthermore, RCTs in this field need better strategies to monitor and enhance adherence and a long duration of follow-up appropriate for cancer outcomes, which can pose serious challenges for their conduct[16,38].

Another important issue is that observational studies in nutritional epidemiology have largely relied on food frequency questionnaires (FFQs) to measure the consumption of the different dietary factors (most studies are in European-descent populations) and a small percentage of studies have included other methods such as 24-h recalls usually in combination with an FFQ. However, this approach is prone to measurement error, especially in the case of assessing intake of non-habitually consumed items (e.g., red meat, citrus fruit), which in prospective studies usually results in risk estimates attenuated toward the null and a loss of statistical power[39]. There is also the issue of estimating the intake of dietary factors (e.g., fiber) that not only depends on multiple episodically consumed food items but also on nutrient databases that may further contribute to measurement error. Furthermore, if the statistical models are adjusted for additional factors, such as additional dietary variables, that are also measured imprecisely, the risk estimates may become attenuated, inflated, or can even change direction[39,40].

Several methods have been proposed to reduce measurement issues in assessing dietary intake. The first approach includes the use of reference methods, such as the 24-h recall[41]. However, multiple measurements, up to 6 days for less frequently consumed foods, are needed to get a more precise and correct assessment of an individual's usual intake[41,42]. The use of mobile phone pictures of all foods consumed and image recognition software for the analysis regarding type of food and amount could be a useful future tool, but this method is still under development[43]. The second option is the use of biomarkers of intake either directly in the analysis or as a calibration tool for the self-reported assessments[44,45]. Unfortunately, there is currently a limited number of such biomarkers, and therefore there is a big need for additional biomarkers to be identified, probably through high-dimensional metabolomic profiling or other omics platforms[46]. Calibration of the biomarker to the true intake would be also required within a feeding study to obtain unbiased estimates[45,47].

Finally, the statistical tests we used to assess bias do not prove its definitive presence or its exact source. However, our estimates are likely to be conservative, as a negative test does not exclude the potential for bias.

**Summary and implications**. The association between diet and risk of cancer has been extensively studied. Taking into account the inclusion of only observational studies and the limitations of the dietary assessment methods that may bias risk estimates, we found strong or highly suggestive evidence to support: (a) the positive association of alcohol consumption and risk of colon, rectum, breast, esophageal, head and neck, and liver cancer, (b) the inverse association of calcium, dairy, and whole grain consumption and risk of CRC, and (c) the inverse association of coffee consumption and risk of liver and skin basal cell carcinoma. Other associations could be genuine, but substantial uncertainty remains. Additional similar research is unlikely to change current evidence for most associations with few exceptions that pertained mostly to currently observed null associations between single dietary factors and understudied malignancies. Future research should instead focus on new and improved methods (e.g., repeated web-based dietary records, biomarkers of nutritional status) to measure the time-varying nature of nutrition, the role of early life diet, the assessment of overall diet patterns, the investigation of the biological processes involved in the diet–cancer associations, the study of molecular cancer subtypes and outcomes after cancer diagnosis, and the interaction of diet patterns with the rest of the exposome (e.g., environment, behavior, genome, metabolome, proteome, epigenome, gut microflora, etc.). For public health and policy, efforts should be targeted to deter the known major diet-related risk factors for cancer, particularly obesity and alcohol consumption.

## Methods

**Data extraction**. Data was extracted from the WCRF Third Expert Report[2], which is one of the most rigorous and systematic analysis of the scientific research on the associations of diet, nutrition, adiposity, and physical activity with risk of cancer development and survival. We focused on meta-analyses investigating the association of all diet-related exposures and risk of cancer development or death at 11 anatomical sites (i.e., head and neck [mouth, pharynx, larynx], esophageal, stomach, CRC, liver, gallbladder, lung, skin [any type, including melanoma, basal call, and squamous cell carcinomas], breast [female], kidney, and urinary bladder), the meta-analyses of which were conducted by WCRF CUP since 2015. The WCRF systematic review for cervical cancer was published in 2018 but did not include meta-analyses for dietary variables and was thus excluded from the current paper[48]. The relevant umbrella reviews for prostate and endometrial cancer[49] have been published separately[50], and WCRF meta-analyses for pancreatic and ovarian cancers were conducted in 2011 and 2013[51,52] and were considered outdated and not included in the current assessment.

The literature search for the WCRF CUP reports was conducted in MEDLINE, and primarily prospective cohort studies, which are considered to represent the highest level of observational evidence, were included in the meta-analyses[2]. Case–control studies were included only in the meta-analyses for head and neck cancer[53,54]. RCTs were retrieved, but they were too few and the assessed interventions were usually irrelevant to specific dietary intakes; therefore, they were not included in the WCRF CUP meta-analyses. More details on the literature search strategy, search terms, and inclusion/exclusion criteria of the WCRF CUP reports are provided in Supplementary Information.

For all the meta-analyses present in the WCRF reports, we extracted the following information at the meta-analysis level: dietary factor, contrast of comparison, and cancer outcome. At the individual study level, we extracted the first author's surname, year of publication, sex, number of cases and total cohort size (or controls), study design, RRs, and 95% CIs. When this information was not available in the WCRF CUP meta-analyses, it was extracted from the individual studies. Eight authors (N.P., E.C., C.K., S.A., M.Z., J.C.K., V.K., and A.P.) performed the data extraction, which was verified independently by four authors (N.P., G.M., A.K., and E.C.) and any disagreements were resolved by discussion.

The appraisal of the evidence was based primarily on comparisons with a continuous exposure contrast, as it was considered a more standardized way of presenting and synthesizing estimates from individual studies. For associations

where the WCRF CUP reports provided results only in the form of top versus bottom or use versus no use comparisons, we used this information instead.

## Data analysis

*Estimation of summary effects and heterogeneity.* We used a random-effects model to estimate the summary associations and the 95% CIs for each dietary factor and cancer comparison[55]. The between-study heterogeneity was quantified with the $I^2$ statistic[56]. To take further into account the heterogeneity among studies, we also estimated the 95% prediction intervals, representing the range in which the effect estimate from a future study addressing the same association is expected to lie[57]. Substantial inconsistency could reflect either genuine heterogeneity between studies or bias.

*Assessment of small study effects and excess significance bias.* We evaluated whether there was an indication of inflated summary estimates due to small study effects[58], by applying the Egger's regression asymmetry test[59]. We considered as an indication of small study effects when the Egger's P value was ≤0.10 and the point estimate of the largest study (smallest standard error) in the meta-analysis was smaller in magnitude than the summary estimate. We examined whether the observed number of studies with nominally statistically significant results (P value <0.05) within a meta-analysis differed from the expected[60]. The expected number of significant results in each meta-analysis was estimated from the sum of the statistical power estimates for each individual study, using an algorithm from a non-central $t$ distribution and the estimate of the largest study in each meta-analysis as the plausible effect size for the tested association[61]. A P value ≤0.10 indicated evidence of excess significance.

*Grading of the evidence.* Meta-analyses showing nominally significant associations were categorized into four evidence groups, namely, strong, highly suggestive, suggestive, and weak evidence, as previously detailed[62,63]. Briefly, associations were considered to be supported by strong evidence if all the following criteria were met: the meta-analysis included >1000 cancer cases, a threshold that provides 80% power for hazard ratios ≥1.20 ($\alpha = 0.05$, Supplementary Fig. 1); the random-effects model had a P value ≤$10^{-6}$ (under the assumption that the tests of statistical significance of the effect in each meta-analysis were valid), a threshold that might substantially reduce false positive findings in a field where low prior probability exists that any single food/nutrient is associated with cancer[64–66]; absence of high heterogeneity ($I^2 < 50\%$); 95% prediction intervals excluded the null value; no evidence of small study effects and excess significance bias. Associations supported by highly suggestive evidence had meta-analyses with >1000 cases; a random-effects P value ≤$10^{-6}$, and the largest study in the meta-analysis was nominally statistically significant. Associations based on meta-analyses with more than 1,000 cases and a random-effects P-value ≤$10^{-3}$[64–66] were categorized as suggestive evidence. The remaining nominally significant associations were graded as weak evidence. We further evaluated the percentage of individual studies in each major (not for food subgroups and cancer subtypes) meta-analysis that adjusted for age, sex (except for breast cancer), smoking, adiposity, and physical activity.

*Calculation of CP and FSN.* To determine whether or not further similar research in diet and cancer may provide a meaningful contribution to the existing evidence, we adapted research synthesis metrics, the FSN and CP. For the nominally non-statistically significant meta-analyses, we quantified the number of future studies (of average weight as those included in the observed meta-analysis) required to achieve a CP of at least 80% to detect a nominally statistically significant effect size equal to the observed meta-analytic summary estimate assuming that the heterogeneity of the updated meta-analysis did not change[67,68]. We further applied the CP approach using the effect of the largest study in the observed meta-analysis as an alternative estimate. For the nominally statistically significant meta-analyses, we determined the number of future studies of average null effect and average weight needed to detect a non-statistically significant summary estimate by calculating Rosenberg's FSN[69].

All data were collected in MS Office 365 and the statistical analyses were conducted using Stata version 14 and R version 4.03.

**Reporting summary.** Further information on research design is available in the Nature Research Reporting Summary linked to this article.

**Disclaimer.** Where authors are identified as personnel of the International Agency for Research on Cancer/World Health Organization, the authors alone are responsible for the views expressed in this article and they do not necessarily represent the decisions, policy, or views of the International Agency for Research on Cancer/World Health Organization.

## Data availability

The authors declare that all data supporting the findings of this study are available within the paper (and in Supplementary Data 6 file). All data were extracted from the WCRF Third Expert Report (https://www.wcrf.org/diet-and-cancer/). Source data are provided with this paper.

## Code availability

The authors declare that all statistical code supporting the findings of this study is available in the Supplementary Data 7 file. Source data are provided with this paper.

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

## Acknowledgements

The authors wish to thank Dr. Teresa Norat and the WCRF CUP team at Imperial College London for their continued efforts in conducting the CUP meta-analyses and their support in retrieving this data for the current umbrella review. The authors wish to thank Dr. Antonia Vlahou, Staff Research Scientist - Professor Level, of the Proteomics Facility at the Biomedical Research Foundation of the Academy of Athens for funding support of E.C. and C.K. This work was supported by the World Cancer Research Fund International Regular Grant Programme (WCRF 2014/1180 to K.K.T.). E.C. was supported by the Hellenic State Scholarships Foundation scholarship funded under the Action "Reinforcement of Postdoctoral Researchers" of the Framework Programme "Development of Human Resources, Education, and Life-long Learning", with priority actions 6, 8, and 9, and co-funded by the European Social Fund and the Hellenic Republic. E.A.P. and M.T.M. were supported by the US NIH (P30 CA006973, T32 CA009314). The study sponsor had no role in the design and conduct of the study; collection, management, analysis and interpretation of the data; preparation, review or approval of the article; and decision to submit the article for publication.

## Author contributions

The study was conceived and designed by K.K.T. The data were acquired and collated by N.P., G.M., E.C., S.A., V.K., J.C.K., C.K., A.P., M.Z., and K.K.T. and analyzed by N.P., G.M., and A.K. The manuscript was drafted by N.P., G.M., A.K., and K.K.T. and revised critically for important intellectual content by E.C., S.A., V.K., J.C.K., C.K., A.P., M.Z., D.S.L., D.S.M.C., M.K., E.N., A.J.C., M.T.M., E.A.P., and M.J.G. All authors gave final approval of the version to be published and have contributed to the manuscript.

## Competing interests
