## [Peer Review File · Nature Communications]

REVIEWER COMMENTS

Reviewer #1 (Remarks to the Author); expert on umbrella meta-analysis:

Dear authors,

This work is an umbrella review about diet/nutrient factors and cancer risk. The authors found associations with primarily alcohol intake and different cancers, as well as an inverse risk relation for coffee with liver and skin basal cell carcinoma, as well as for calcium, dairy and wholegrain consumption and the risk of CRC.

Main weakness:

The risk association of alcohol consumption and cancers, such as colorectal, esophageal, head & neck and liver cancer is well known. The associations of dairy products, milk, calcium and wholegrains and the lower CRC risk is less established, but this work does not add substantial new insights beyond already published data, similar as with the inverse risk for cancer and coffee consumption.

The innovative part is an analyses concerning the estimated number of additional studies to detect potential relevant/significant results.

Major issues:

This review includes only observational studies, even though the authors focused on prospective studies. This is prone to the fact, that very few RCTs are available in this field, but to confirm suspected benefits due to nutrients, RCTs would be essential. I would encourage to include this as an essential part of the conclusion drawn. In addition, as mentioned by the authors, it is very difficult to assess nutrition during the course of disease or even prior of its occurrence and the data quality is limited.

More or less all included studies were part of the WCRF reports since 2015.

There is no detailed literature research strategy given and no numbers/details concerning overall found publications and MedLine was the only searched database.

A clear literature research strategy and inclusion/exclusion criteria and details for each publication would increase the value of this work. Also removal of duplicate data would be of interest.

There is no clear quality assessment of the meta-analyses included in this article, even though they investigated small study effects and heterogeneity for the cohorts.

The data given in the results part might be better reported in tables as it is hard to read and no further information beyond numbers is given. I would further recommend for the written part another structure, focussing on the diet/nutrient or cancer entity – summarizing all relevant results under this structure.

Within the discussion for potential relevant nutrients, details on mechanisms of actions should be included and also a discussion concerning the impact of nutrients in comparison to other well-established risk factors, such as smoking and viral-exposures, f.e. would be of interest.

If possible, it would be great to establish, if nutrients are independent risk factors beyond smoking habits, obesity and lack of physical activity and others, or if a general life style might be responsible for higher or lower cancer risks.

I would suggest to add these analyses to an umbrella review of the field.

Minor points:

The lines 72-78 within the introduction lack a direct association with the article. I would recommend to leave this passage, place it at the discussion or shorten it significantly.

In total this article might benefit from a clear hypothesis, literature search strategy and presentation of results, as several parts are confusing and hard to read.

Please, cover the extensive literature in the field in the introduction and discussion, as would be expected in a review of this journal.

Reviewer #2 (Remarks to the Author); expert on epidemiology, nutrition and cancer:

Drs. Papadimitriou, Tsilidis and colleagues have assembled a voluminous and comprehensive "umbrella" review of dietary factors and 11 cancers using WCRF report #3 and other data. Several statistical/quantitative approaches to the vetting of published data are used to reach fairly conservative conclusions regarding etiological associations. It is not surprising that many if not most of the highlighted associations mirror those presented in the WCRF report. The primary data tables and figures are clear and appropriate, although for Figure 1 the left and right y-axis labels are not easily interpretable unless they are intended to be one and the same. The supplemental data are important and should be included.

I have only a few specific questions and comments intended to improve interpretation.

1.

It might be going a bit out on a limb, but some comment about whether stronger findings for alcohol, coffee and dairy result from better/more accurate questionnaire capture of these compared to other foods/nutrients or is purely biologically based in cancer etiology.

2.

As the authors mention under study strengths/limitations, analysis and presentation of the CP and FSN metrics could be construed as discouraging thresholds for future research and studies. Some greater discussion of these, including likely fruitful areas of research, would be helpful to readers reflecting on gaps and next steps in the field.

3.

Some mention of when similar updating of meta-analyses for Pancreas and Ovary (which were considered as too dated within the WCRF report) would be useful.

Reviewer #3 (Remarks to the Author); expert on epidemiology, nutrition and cancer:

In the article “Diet and cancer risk at 11 anatomical sites: an umbrella review of the evidence”, Papadimitriou et al clearly and concisely summarize an expansive body of literature. Their exhaustive, high-level, quantitative approach is a major strength, but, to a lesser extent, it is also a limitation. More specifically, I think that by considering numerous dietary factors and cancer sites simultaneously, they accomplished their stated goal of identifying the most robust diet-cancer associations, but it is, to some degree, at the expense of the details, which are critical to understanding the limitations of the current body of literature as well as how the field can move forward.

For example, in the section “Conclusions and implications” the authors clearly and succinctly communicate what is needed to advance epidemiological studies of diet and cancer (e.g., improved and serial dietary assessment, biomarker development, etc.) and what is not (i.e., more of the same). However, I don’t think that the abstract or introduction provide the context for this important discussion. The significance of the word “similar” in the statement “similar research was unlikely to change this evidence” in the abstract or the statement that measurement error is an issue in nutritional epidemiology in the introduction is likely to be lost on many readers. I would suggest a brief discussion of how nutritional epidemiology studies, to date, have largely relied on food frequency questionnaires and that their search for “robust” associations is in this context. Additionally, it may be instructive to discuss that not all dietary factors are equally well-captured by FFQ. For example, FFQs may capture habitually consumed items like coffee with less error than more episodically consumed items like red meat or citrus fruit. There is also the issue of estimating intake of dietary factors (e.g., fiber) that not only depend on multiple episodically consumed food items but also on nutrient databases that may further contribute to measurement error. I realize that a study by study or even association by association discussion of the potential sources of error is not tenable in an umbrella review with such a broad scope. However, a brief overview of the sources of error that potentially impact the included meta-analyses would provide needed context and would help to support the later discussion of how the field can move forward.

Overall, I think this is an impressive effort to make sense of hundreds of meta-analyses that likely cover thousands of studies. I have several additional questions and comments for the authors’ consideration:

- The authors do a good job of highlighting high heterogeneity as an important factor for objectively assessing the strength of the evidence, but they do not provide the reader with sufficient background to understand what the presence or absence of high heterogeneity indicates. As noted in the results, head and neck cancer had the largest proportion of high heterogeneity, but why? Is this because meta-analyses of diet and head and neck cancer were the only ones to include case-control-control studies? At the other end of the spectrum, no meta-analyses of gallbladder cancer had high heterogeneity, but there were only two meta-analyses, which included only a few studies. Regardless of cancer site, was heterogeneity higher for certain dietary factors than others or for meta-analyses that included more geographically diverse studies? What does this tell us about potential sources of bias or generalizability?
- The rationale for which cancer sites were included in the review was clearly stated, but I did not find a similar rationale for which dietary factors were included. Please clarify.
- How were meta-analysis that included the same studies handled? Is it possible that some

studies are included multiple times in summary estimates?

- I see that the sample dataset provided includes “Author_year” as well as exposure and outcome data so that one could trace back to the original meta-analysis. Will the complete dataset be made available so that it is possible to know which meta-analyses were included for each diet-cancer association? As far as I can tell, there is presently no straightforward way to do this.
- It would be helpful if Supplement table 2 was available in an excel format, in which the column headers were frozen, and the data could be sorted by outcome, exposure, etc.
- For rare cancers were there any instances in which the evidence was graded as “weak” because there were fewer than 1000 cases in the meta-analyses even though all other criteria were met?
- What do the blue boxes in Supplement Table 3 indicate?

Reviewer #4 (Remarks to the Author); expert on biostatistics:

1. This is a comprehensive review of 860 meta-analyses of observational studies with the purpose to evaluate the strength and validity of the evidence for association between food and nutrient intakes and risk of developing or dying from eleven different primary cancers. The authors acknowledge in the Introduction (p.3, lines 67-71) that “reported meta-analytic estimates from observational studies might not represent causal associations but may emerge due to biases common across studies, such as exposure measurement error, residual confounding and publication bias, subsequently diminishing the strength of aggregated scientific evidence”. The authors’ motivation therefore was to “systematically evaluate the robustness of the observational meta-analytic evidence across a large number of food and nutrient associations with risk of cancer at 11 anatomical sites.” The authors applied several criteria to evaluate the strength and validity of the evidence for diet-cancer associations, including categorizing meta analyses with nominally significant results into four evidence groups representing strong, highly suggestive, suggestive, and weak evidence based on the number of cases, p-values in the random effects meta-analytic models, degree of heterogeneity, whether the 95% prediction intervals excluded the null value, evidence of small study effects, and excess significance bias. They also evaluated whether additional research is or is not warranted to change the inferences from the existing meta-analyses using an adaptation of research synthesis methods. Unfortunately, this rather sophisticated evaluation may be prone to misinterpreted and even spurious results as I explain below.

2. Before providing my major concerns, I would like to make a comment on categorization of evidence into 4 groups. The first criterion listed by authors (p. 9, last paragraph), the number of cases in the meta-analysis, defines the power to detect the effect of dietary intake of interest in each of 860 meta-analyses. The second criterion, the p-value in the random effects meta-analytic model, defines the significance level for each meta-analysis. For the moment, assuming that the tests of statistical significance of the effect in each meta-analysis were valid, i.e., based on the correctly formulated null and alternative hypotheses, the fact that there were 860 such tests requires some adjustment for multiplicity to control the

number of false positive (overall significance level) as well as false negative results (related to overall power). For strong evidence of the existing relationship, the authors used a p-value less or equal to 10^{-6} . This looks like an approximate Bonferroni adjustment to control the experiment-wise significance level at 5% (the more precise would be a level of $1 - (0.95)^{1/860} = 6 \times 10^{-5}$). Given this, I do not understand using a p-value of 10^{-3} for a suggestive evidence, because the corresponding experiment-wise significance level would be controlled at $1 - 0.999860 = 57.7\%$, hardly an appropriate level for suggestive evidence. Given the conservative nature of Bonferroni adjustment and many simultaneous tests, it may be advantageous to control the false discovery rate known as FDR at 5% or even 10%. Now, multiplicity requires also to appropriately adjust the overall power. The required by authors at least 1,000 cases for strong to suggestive evidence is not explained and not translated to this power. A properly adjusted for multiplicity power should be presented to make sure that 1,000 required cases were not based on underpowered studies. As I mentioned, the required multiplicity adjustment still assumes that the tests were valid. Unfortunately, this may not be a correct assumption, as I explain in the next comment.

3. I will concentrate on one very specific problem in nutritional epidemiology, namely measurement error in dietary assessment methods. Measurement error is mentioned in the introduction (p.3) as being present in dietary self-report (the basis of diet evaluation in nutritional studies) but then is never discussed again in the manuscript. The details of the effects of exposure measurement error on estimated relative risks are not provided aside from mentioning that error may produce biased results. Yet, based on recent evidence from validation studies with the so-called recovery (unbiased) biomarkers of several dietary intakes (1 - 4), it is well recognized that dietary measurement error may be substantial and include both random and systematic components. Moreover, it is probably larger than that for most other exposures of common epidemiological interest. Even nondifferential measurement error (i.e., error that is unrelated to the outcome given true exposure), which is usually hypothesized in cohort studies, creates three problems: 1) bias in estimated relative risks; 2) loss of statistical power to detect diet–disease relationships; and 3) in models with two or more dietary components, invalidity of the conventional statistical tests and confidence intervals for detecting those relationships. It is well known in the statistical literature that measurement error in a risk model with only one covariate leads to a multiplicative bias in an estimated association. In dietary studies, this association is usually attenuated, i.e., biased toward the null. The corresponding statistical test loses power but remains valid. On the other hand, in a multivariate model with two or more mismeasured exposures, estimated covariate effects may become attenuated, inflated, or can even change direction, and consequently their tests not just lose power but become invalid. Thus, one cannot tell whether a nominally statistically significant effect indicates a real association. Similarly, the calculated confidence intervals do not correspond to the nominal confidence level. Unfortunately, this will also happen to a predictive interval that the authors use in categorizing meta-analytic evidence. The described problem is important because investigators almost always include more than one nutritional exposure in disease models. This change in nature of the bias in the estimated effect in models with two or more dietary exposures arises from a phenomenon known as residual confounding caused by measurement error. When there are several nutritional intake covariates that are mismeasured, each will adopt a part of the effect of the others, and the fractions of the effects that are adopted will depend on the relative sizes of the errors and the correlations among them. A meta-analysis of studies with similar structures of measurement error may increase the detrimental effects of residual confounding. In the literature, several investigators have cautioned against overinterpreting apparently highly precise results

reported from meta-analyses of observational studies. For example, Egger et al. (6) warn that residual confounding can distort findings from observational studies and of the consequent “danger that meta-analyses of observational data produce very precise but equally spurious results.” They conclude that “the statistical combination of data should not therefore be a prominent component of reviews of observational studies.” This concern was expressed for all observational studies where residual confounding may be caused by unknown or unobserved covariates. In nutritional epidemiology, the possibility of residual confounding becomes especially acute due to substantial measurement error (7). It is therefore critical that the present authors carefully discuss effects of dietary measurement error and how it may change the interpretation of their evaluation.

4. Following up on my previous comment, the structure of dietary measurement error depends on both the dietary assessment instrument and the study population to which this instrument is applied. It is therefore important that the authors provide information on the dietary assessment instruments involved in their meta-analyzed studies and the corresponding populations. It would also be of interest if the authors could discuss some suggested ideas of how to mitigate the detrimental effects of dietary measurement error, for example, provided in (7 - 8).

References

1. Kipnis V, Subar AF, Midthune D, Freedman LS, Ballard-Barbash R, Troiano RP, Bingham S, Schoeller DA, Schatzkin A, Carroll RJ. The structure of dietary measurement error: results of the OPEN biomarker study. *Am J Epidemiol* 2003;158:14-21.
2. Kipnis V, Freedman LS. Impact of exposure measurement error in nutritional epidemiology. *J Natl Cancer Inst* 2008; 100:1658-9.
3. Freedman LS, Moler J, Commins J, Arab L, Baer D, Kipnis V, Midthune D, Moshfegh A, Neuhauser ML, Prentice R, Schatzkin A, Spiegelman D, Subar A, Tinker L, Willett W. Pooled results from five validation studies of dietary self-report instruments using recovery biomarkers for energy and protein intake, *Am J Epidemiol* 2014;180,172-88.
4. Freedman LS, Commins JM, Moler JE, Willett W, Tinker LF, Subar AF, Spiegelman D, Rhodes D, Potischman N, Neuhauser ML, Moshfegh AJ, Kipnis V, Arab L, Prentice RL. Pooled results from 5 validation studies of dietary self-report instruments using recovery biomarkers for potassium and sodium intake. *Am J Epidemiol*. 2015 Apr 1;181(7):473-87.
5. Freedman LS, Commins JM, Willett W, Tinker LF, Spiegelman D, Rhodes D, Potischman N, Neuhauser ML, Moshfegh AJ, Kipnis V, Baer DJ, Arab L, Prentice RL, and Subar AF. Evaluation of the 24-Hour Recall as a Reference Instrument for Calibrating Other Self-Report Instruments in Nutritional Cohort Studies: Evidence from the Validation Studies Pooling Project. *Am J Epidemiol* 2017;186(1),73-82.
6. Egger M, Schneider M, Davey-Smith G. Meta-analysis spurious precision? Meta-analysis of observational studies. *Br Med J*. 1998;316(7125):140–144.
7. Prentice RL, Huang Y. Nutritional epidemiology methods and related statistical challenges and opportunities. *Stat Theory Relat Fields* 2018;2(1):2-10.
8. Prentice RL. Dietary assessment opportunities to enhance nutritional epidemiology evidence. *Ann Intern Med* 2020;172:354-355.

Victor Kipnis

Reply to reviewer comments

Reviewer comments

Reviewer #1

1) The risk association of alcohol consumption and cancers, such as colorectal, esophageal, head & neck and liver cancer is well known. The associations of dairy products, milk, calcium and wholegrains and the lower CRC risk is less established, but this work does not add substantial new insights beyond already published data, similar as with the inverse risk for cancer and coffee consumption. The innovative part is an analyses concerning the estimated number of additional studies to detect potential relevant/significant results.

Reply: The reviewer is right that some of the associations described in this paper have been reported extensively previously, but the aim of the current study was to systematically evaluate the robustness of these associations across a large number of foods and nutrients in relation to risk of cancer at 11 anatomical sites, which is novel. With a rapidly evolving evidence base and lack of a robust synthesis of the published literature, questions regarding the association of diet with cancer are becoming increasingly difficult to answer. We further evaluated whether additional research is or is not warranted to change the inferences from the existing meta-analyses using an adaptation of research synthesis methods, and offered recommendations where future research in the nutritional epidemiology of cancer should focus. Please read below for specific substantive revisions that were added in this work to further increase its rigor.

2) This review includes only observational studies, even though the authors focused on prospective studies. This is prone to the fact, that very few RCTs are available in this field, but to confirm suspected benefits due to nutrients, RCTs would be essential. I would encourage to include this as an essential part of the conclusion drawn. In addition, as mentioned by the authors, it is very difficult to assess nutrition during the course of disease or even prior of its occurrence and the data quality is limited. More or less all included studies were part of the WCRF reports since 2015.

Reply: We thank the reviewer for allowing us to expand on these points. We have added several paragraphs in the Discussion (lines: 383-390) to acknowledge the advantages and limitations of RCTs in the field of nutritional epidemiology, and to discuss further the limitations of observational studies in this field prioritizing the issue of exposure misclassification (lines: 391-404) and how future studies could ameliorate the assessment of diet (lines: 405-418). We have rephrased our conclusion statement to better acknowledge the latter issues, as:

“Taking into account the inclusion of only observational studies and the limitations of the dietary assessment methods, we found strong or highly suggestive evidence...”

The reviewer is right that we used studies identified in the WCRF 3rd expert report that was published in 2018, but we consider this to be an advantage of our approach, as this report is one of the most rigorous and systematic analysis of the scientific research on the associations of diet, nutrition, adiposity and physical activity with risk of cancer development. Our study went beyond the WCRF report to systematically evaluate the robustness of these associations using an independent methodology, evaluate research gaps and suggest recommendations for future research.

3) *There is no detailed literature research strategy given and no numbers/details concerning overall found publications and MedLine was the only searched database. A clear literature research strategy and inclusion/exclusion criteria and details for each publication would increase the value of this work. Also removal of duplicate data would be of interest.*

Reply: We thank the reviewer for this comment. More details on the literature search strategy, search terms and inclusion/exclusion criteria of the WCRF CUP reports are now provided in Supplementary Methods.

4) *There is no clear quality assessment of the meta-analyses included in this article, even though they investigated small study effects and heterogeneity for the cohorts.*

Reply: We did not perform quality assessment of the meta-analyses using one of the available mechanistic risk of bias assessments, because such instruments prioritize the repeatability of the process over a more thoughtful and informative but necessarily somewhat more subjective approach (Savitz DA, et al. Am J Epidemiol 2019;188:1581-1585). In addition, all included meta-analyses were performed by the WCRF CUP team using same research protocols (please see Supplementary Methods). We did however assess evidence for presence or absence of between study heterogeneity, small study effects and excess significance bias.

5) *The data given in the results part might be better reported in tables as it is hard to read and no further information beyond numbers is given. I would further recommend for the written part another structure, focussing on the diet/nutrient or cancer entity – summarizing all relevant results under this structure.*

Reply: This umbrella review summarizes and evaluates the evidence from a large literature (N=860 meta-analyses), and we have opted for a more succinct description of the findings in the Results. Detailed findings by diet-cancer association are provided in supplementary tables. However, we have now revised the Results section to include more information on specific dietary variables and diet-cancer associations. In addition, we have added two new supplementary tables (Supplementary tables 3 and 4) that summarize the evidence by diet exposure categories.

6) *Within the discussion for potential relevant nutrients, details on mechanisms of actions should be included and also a discussion concerning the impact of nutrients in comparison to other well-established risk factors, such as smoking and viral-exposures, f.e. would be of interest. If possible, it would be great to establish, if nutrients are independent risk factors beyond smoking habits, obesity and lack of physical activity and others, or if a general life style might be responsible for higher or lower cancer risks. I would suggest to add these analyses to an umbrella review of the field.*

Reply: We have now added several paragraphs in the Discussion (lines: 312-358), which review the potential biological mechanisms involved in diet-cancer associations that received high evidence grade for an association in the current umbrella review. In addition, we evaluated the percentage of individual studies in each major (not for food subgroups and cancer subtypes) meta-analysis that adjusted for age, sex (except for breast cancer), smoking, adiposity and physical activity, and added the findings in Supplement Table 2 and reported in Results. Among the associations supported by strong or highly suggestive evidence, the majority of individual studies in each respective meta-analysis had adjusted for smoking (67%-100%) and adiposity (67%-95%), but adjustment for physical activity was less

frequently conducted (5%-79%). Adjustments for viral agents are in general not performed, as this information is usually missing from large well-established cohorts.

7) The lines 72-78 within the introduction lack a direct association with the article. I would recommend to leave this passage, place it at the discussion or shorten it significantly.

Reply: These lines have now been removed.

8) In total this article might benefit from a clear hypothesis, literature search strategy and presentation of results, as several parts are confusing and hard to read.

Reply: Please read replies to previous relevant comments.

9) Please, cover the extensive literature in the field in the introduction and discussion, as would be expected in a review of this journal.

Reply: We have expanded substantially the Discussion (as reported in more detail in previous relevant comments) to include current evidence and mechanisms of several diet-cancer associations, main limitations of this research field and potential future fixes.

Reviewer #2

1) Drs. Papadimitriou, Tsilidis and colleagues have assembled a voluminous and comprehensive "umbrella" review of dietary factors and 11 cancers using WCRF report #3 and other data. Several statistical/quantitative approaches to the vetting of published data are used to reach fairly conservative conclusions regarding etiological associations. It is not surprising that many if not most of the highlighted associations mirror those presented in the WCRF report. The primary data tables and figures are clear and appropriate, although for Figure 1 the left and right y-axis labels are not easily interpretable unless they are intended to be one and the same. The supplemental data are important and should be included.

Reply: We thank the reviewer for the positive comments. The labels have now been improved in Figure 1.

2) It might be going a bit out on a limb, but some comment about whether stronger findings for alcohol, coffee and dairy result from better/more accurate questionnaire capture of these compared to other foods/nutrients or is purely biologically based in cancer etiology.

Reply: The reviewer is right that not all dietary factors are equally well-captured by diet assessment questionnaires. We have now expanded the limitations section in the Discussion to describe this phenomenon (lines: 391-404). It is difficult to judge whether stronger findings observed for alcohol, coffee, dairy products and cancer risk are due to a more accurate capture of these intakes or is purely biological. It could be both mechanisms playing a role. The meta-analyses investigating associations on alcohol and cancer risk had among the highest between-study heterogeneity compared to other diet groups, which could reflect true heterogeneity but also different levels of exposure measurement error between studies.

3) As the authors mention under study strengths/limitations, analysis and presentation of the CP and FSN metrics could be construed as discouraging thresholds for future research and studies. Some greater discussion of these, including likely fruitful areas of research, would be helpful to readers reflecting on gaps and next steps in the field.

Reply: Thank you! Our evaluation of the literature using the CP and FSN metrics have showed that additional similar research is unlikely to change current evidence for most diet-cancer associations with few exceptions that pertained mostly to currently observed null associations between single dietary factors and understudied malignancies. We have tried to communicate throughout the paper what is needed to advance epidemiological studies of diet and cancer (e.g., improved and serial dietary assessment, biomarker development) and what is not (i.e., more of the same). We expanded the Discussion (lines: 405-418) to include more detailed suggestions for future research to reduce measurement issues in dietary assessment. In addition, we suggested in the conclusion which new areas or research questions should be prioritized in the future (e.g., role of early life diet, assessment of overall diet patterns, investigation of the biological processes involved in the diet-cancer associations, study of molecular cancer subtypes and outcomes after cancer diagnosis, and the interaction of diet patterns with the rest of the exposome).

4) Some mention of when similar updating of meta-analyses for Pancreas and Ovary (which were considered as too dated within the WCRF report) would be useful.

Reply: Updated meta-analyses for pancreatic and ovarian cancer risk are warranted, but they are not prioritized by the WCRF CUP team for the next one or two years.

Reviewer #3

1) In the article “Diet and cancer risk at 11 anatomical sites: an umbrella review of the evidence”, Papadimitriou et al clearly and concisely summarize an expansive body of literature. Their exhaustive, high-level, quantitative approach is a major strength, but, to a lesser extent, it is also a limitation. More specifically, I think that by considering numerous dietary factors and cancer sites simultaneously, they accomplished their stated goal of identifying the most robust diet-cancer associations, but it is, to some degree, at the expense of the details, which are critical to understanding the limitations of the current body of literature as well as how the field can move forward.

Reply: We thank the reviewer for this comment! We have now expanded the manuscript substantially to better describe details and limitations of the current body of literature. Specific changes are highlighted in the reply to the subsequent comments.

2) For example, in the section “Conclusions and implications” the authors clearly and succinctly communicate what is needed to advance epidemiological studies of diet and cancer (e.g., improved and serial dietary assessment, biomarker development, etc.) and what is not (i.e., more of the same). However, I don’t think that the abstract or introduction provide the context for this important discussion. The significance of the word “similar” in the statement “similar research was unlikely to change this evidence” in the abstract or the statement that measurement error is an issue in nutritional epidemiology in the introduction is likely to be lost on many readers. I would suggest a brief discussion of how nutritional epidemiology studies, to date, have largely relied on food frequency questionnaires and that their search for “robust” associations is in this context. Additionally, it may be instructive to discuss that not all dietary factors are equally well-captured by FFQ. For example, FFQs may capture habitually consumed items like coffee with less error than more episodically consumed items like red meat or citrus fruit. There is also the issue of estimating intake of dietary factors (e.g., fiber) that not only depend on multiple episodically consumed food items but also on nutrient databases that may further contribute to measurement error. I realize that a study by study or even association by association discussion of the potential sources of error is not tenable in an umbrella review with such a broad scope. However, a brief overview of the sources of error that potentially impact the included meta-analyses would provide needed context and would help to support the later discussion of how the field can move forward.

Reply: We thank the reviewer for this important comment! We have now included a new paragraph in the Discussion (lines: 391-404) to better describe the sources of error in nutritional epidemiology studies, and a separate paragraph (lines: 405-418) for suggestions to reduce measurement issues in assessing dietary intake. We have further revised the Abstract and Introduction to better provide context of these measurement issues earlier on in the manuscript.

3) The authors do a good job of highlighting high heterogeneity as an important factor for objectively assessing the strength of the evidence, but they do not provide the reader with sufficient background to understand what the presence or absence of high heterogeneity indicates. As noted in the results, head and neck cancer had the largest proportion of high heterogeneity, but why? Is this because meta-analyses of diet and head and neck cancer were the only ones to include case-control-control studies? At the other end of the spectrum, no meta-analyses of gallbladder cancer had high heterogeneity, but there were only two meta-analyses, which included only a few studies. Regardless of cancer site, was heterogeneity higher for certain dietary factors than others or for meta-analyses that included more geographically diverse studies? What does this tell us about potential sources of bias

or generalizability?

Reply: We have now analyzed and described in the Results the summary characteristics of the included meta-analyses also by type of dietary exposure, as outlined in detail below. This information is added in two new supplementary tables (Supplementary Tables 3 and 4).

Associations for fruits/vegetables and alcohol had the largest number of meta-analyses (N=184 and 141, respectively) followed by meat/eggs and beverages (N=118 and 70, respectively). The median number of individual studies within a meta-analysis was 5 and was similar for different dietary exposures with the exception of fats/fatty acids, where the median was 11. The overall median number of cancer cases in a meta-analysis was 2,152 and ranged from 1,038 for phytochemicals to 9,955 for fiber. The following comment was added in the Discussion: *“Overall, a large number of meta-analyses (N=860) was evaluated, but they contained on average relatively few studies (median=5) and moderate number of cancer cases (median=2,152).”*

The percentage of nominally significant meta-analyses ranged from 0% for legumes/soy products to 54% for alcohol and 67% for grains (which had only three available meta-analyses). Associations for energy/sugars and vitamin D had the largest proportion of meta-analyses with high heterogeneity (50% each) followed by alcohol (43%). Sixty-nine (8%) meta-analyses showed evidence of small study effects bias, which was highest for salt/salty foods (25%; only four available meta-analyses) followed by alcohol (16%) and fruits/vegetables (13%). Additionally, the proportion of meta-analyses showing evidence of excess significance bias was highest for associations that involved alcohol (35%) and salt/salty foods (50%). The following comment was added in the Discussion: *...“The latter summary description of the robustness of the evidence seems to apply for most dietary exposure categories and cancer outcomes, with one notable exception. A larger proportion (11%) of alcohol and cancer associations (compared to other diet-cancer associations) is supported by strong statistical evidence, but potential evidence for existence of bias is also present in the alcohol and cancer associations shown by larger likelihoods of high heterogeneity, small study effects and excess significance bias.”*

We further added in the Methods that substantial heterogeneity could reflect either genuine heterogeneity between studies or bias.

Regarding the large heterogeneity observed for head and neck cancer, we added in the Results, that it was substantially reduced after removing a pooling project of case-control studies.

4) The rationale for which cancer sites were included in the review was clearly stated, but I did not find a similar rationale for which dietary factors were included. Please clarify.

Reply: All dietary factors associated with any of the included cancers in meta-analyses from the relevant WCRF CUP reports were included in this umbrella review. A clarification was added in the Methods.

5) How were meta-analysis that included the same studies handled? Is it possible that some studies are included multiple times in summary estimates?

Reply: No study was included twice in any meta-analysis. This had been now clarified in the Supplementary Methods.

6) I see that the sample dataset provided includes “Author_year” as well as exposure and

outcome data so that one could trace back to the original meta-analysis. Will the complete dataset be made available so that it is possible to know which meta-analyses were included for each diet-cancer association? As far as I can tell, there is presently no straightforward way to do this.

Reply: Yes, we have provided the complete raw data in an Excel file.

7) It would be helpful if Supplement table 2 was available in an excel format, in which the column headers were frozen, and the data could be sorted by outcome, exposure, etc.

Reply: All supplemental tables are now provided in an Excel format.

8) For rare cancers were there any instances in which the evidence was graded as “weak” because there were fewer than 1000 cases in the meta-analyses even though all other criteria were met?

Reply: No, this was not the case for any diet-cancer association.

9) What do the blue boxes in Supplement Table 3 indicate?

Reply: This was a typographical error. It has been corrected.

Reviewer #4

1) This is a comprehensive review of 860 meta-analyses of observational studies with the purpose to evaluate the strength and validity of the evidence for association between food and nutrient intakes and risk of developing or dying from eleven different primary cancers. The authors acknowledge in the Introduction (p.3, lines 67-71) that “reported meta-analytic estimates from observational studies might not represent causal associations but may emerge due to biases common across studies, such as exposure measurement error, residual confounding and publication bias, subsequently diminishing the strength of aggregated scientific evidence”. The authors’ motivation therefore was to “systematically evaluate the robustness of the observational meta-analytic evidence across a large number of food and nutrient associations with risk of cancer at 11 anatomical sites.” The authors applied several criteria to evaluate the strength and validity of the evidence for diet-cancer associations, including categorizing meta analyses with nominally significant results into four evidence groups representing strong, highly suggestive, suggestive, and weak evidence based on the number of cases, p-values in the random effects meta-analytic models, degree of heterogeneity, whether the 95% prediction intervals excluded the null value, evidence of small study effects, and excess significance bias. They also evaluated whether additional research is or is not warranted to change the inferences from the existing meta-analyses using an adaptation of research synthesis methods. Unfortunately, this rather sophisticated evaluation may be prone to misinterpreted and even spurious results as I explain below.

Reply: We thank the reviewer for his comments! We have now expanded the manuscript substantially to better describe details and limitations of the current body of literature. Specific changes are highlighted in the reply to the subsequent comments.

2) Before providing my major concerns, I would like to make a comment on categorization of evidence into 4 groups. The first criterion listed by authors (p. 9, last paragraph), the number of cases in the meta-analysis, defines the power to detect the effect of dietary intake of interest in each of 860 meta-analyses. The second criterion, the p-value in the random effects meta-analytic model, defines the significance level for each meta-analysis. For the moment, assuming that the tests of statistical significance of the effect in each meta-analysis were valid, i.e., based on the correctly formulated null and alternative hypotheses, the fact that there were 860 such tests requires some adjustment for multiplicity to control the number of false positive (overall significance level) as well as false negative results (related to overall power). For strong evidence of the existing relationship, the authors used a p-value less or equal to 10^{-6} . This looks like an approximate Bonferroni adjustment to control the experiment-wise significance level at 5% (the more precise would be a level of $1 - (0.95)^{1/860} = 6 \times 10^{-5}$). Given this, I do not understand using a p-value of 10^{-3} for a suggestive evidence, because the corresponding experiment-wise significance level would be controlled at $1 - 0.999860 = 57.7\%$, hardly an appropriate level for suggestive evidence. Given the conservative nature of Bonferroni adjustment and many simultaneous tests, it may be advantageous to control the false discovery rate known as FDR at 5% or even 10%. Now, multiplicity requires also to appropriately adjust the overall power. The required by authors at least 1,000 cases for strong to suggestive evidence is not explained and not translated to this power. A properly adjusted for multiplicity power should be presented to make sure that 1,000 required cases were not based on underpowered studies. As I mentioned, the required multiplicity adjustment still assumes that the tests were valid. Unfortunately, this may not be a correct assumption, as I explain in the next comment.

Reply: Thank you for the detailed comments! Our aim in this umbrella review was to summarize and appraise the robustness of the evidence for the association between food and nutrient intake and risk of 11 cancers using the most extensive systematic review to date from the WCRF CUP. We applied several statistical criteria and

sensitivity analyses to evaluate the strength and validity of this evidence. Several different methods exist for rating quality of evidence, but they are inconsistent and allow some degree of arbitrariness. The criteria we used for grading evidence should not be considered causal criteria, especially when used individually, but we think that they are useful for identifying biases (and evaluate the robustness of the evidence) when used together. These criteria have been applied before extensively in several fields to evaluate associations between lifestyle/environmental exposures and risk of chronic diseases (e.g., Kyrgiou M, et al. Adiposity and cancer at major anatomical sites: umbrella review of the literature. *BMJ* 2017;356:j477; Rezende LFM, et al. Physical activity and cancer: an umbrella review of the literature including 22 major anatomical sites and 770 000 cancer cases. *British journal of sports medicine* 2018;52:826-33; Bellou V, et al. Risk factors for type 2 diabetes mellitus: An exposure-wide umbrella review of meta-analyses. *PLoS One* 2018;13:e0194127). The latter points have been already included in the Discussion.

One of the statistical criteria we used was the p-value of the random-effects meta-analytic model. A p-value threshold of 10^{-3} was used (among other criteria) to denote suggestive evidence and a lower threshold of 10^{-6} was used to denote highly suggestive and strong evidence. These thresholds were chosen to reduce the probability of false positive findings, but their selection was not strictly based on a specific Bonferroni or FDR calculation to adjust for multiplicity. The logic behind our selection is outlined below. P-values measure the strength of the evidence against the null hypothesis, and the smaller the p-value the stronger the evidence against the null hypothesis. The choice of any particular threshold is arbitrary and involves a trade-off between type I and type II errors, as the reviewer also suggested. We chose the threshold of 0.001 to represent suggestive evidence, because it has been shown to reduce the false positive rate to reasonable levels (i.e., 5%) and also because it corresponds to reasonable large Bayes factors in favour of the alternative hypothesis (Johnson VE. Revised standards for statistical evidence. *PNAS* 2013;110:19313-17; Benjamin DJ, et al. Redefine statistical significance. *Nat Human Behaviour* 2018;2:6-10). For exploratory research with very low prior probabilities of association that could describe to some extent the field of nutritional epidemiology of cancer, lower significance thresholds would be needed and we arbitrarily chose 0.000001 to represent highly suggestive and strong evidence. We rephrased the relevant paragraph in the Methods to clarify our selection of the p-value thresholds.

The reviewer is right that correction for multiplicity requires also to appropriately adjust the overall power, as low statistical power and $\alpha=0.05$ have been shown to produce high false positive rates. We chose a threshold of 1,000 cancer cases after performing power estimations (provided as a new Supplement Figure 1) and observing that such a threshold provides ~80% power for most often observed risk estimates in epidemiology ($HR \geq 1.20$; $\alpha=0.05$). Higher thresholds of cancer cases would be needed to achieve ~80% power when α is lower, but the median number of cancer cases across the included meta-analyses was substantially larger (i.e. 2,152).

3) I will concentrate on one very specific problem in nutritional epidemiology, namely measurement error in dietary assessment methods. Measurement error is mentioned in the introduction (p.3) as being present in dietary self-report (the basis of diet evaluation in nutritional studies) but then is never discussed again in the manuscript. The details of the effects of exposure measurement error on estimated relative risks are not provided aside from mentioning that error may produce biased results. Yet, based on recent evidence from validation studies with the so-called recovery (unbiased) biomarkers of several dietary intakes (1 - 4), it is well recognized that dietary measurement error may be substantial and include both random and systematic components. Moreover, it is probably larger than that for most other exposures of common epidemiological interest. Even nondifferential

measurement error (i.e., error that is unrelated to the outcome given true exposure), which is usually hypothesized in cohort studies, creates three problems: 1) bias in estimated relative risks; 2) loss of statistical power to detect diet–disease relationships; and 3) in models with two or more dietary components, invalidity of the conventional statistical tests and confidence intervals for detecting those relationships. It is well known in the statistical literature that measurement error in a risk model with only one covariate leads to a multiplicative bias in an estimated association. In dietary studies, this association is usually attenuated, i.e., biased toward the null. The corresponding statistical test loses power but remains valid. On the other hand, in a multivariate model with two or more mismeasured exposures, estimated covariate effects may become attenuated, inflated, or can even change direction, and consequently their tests not just lose power but become invalid. Thus, one cannot tell whether a nominally statistically significant effect indicates a real association. Similarly, the calculated confidence intervals do not correspond to the nominal confidence level. Unfortunately, this will also happen to a predictive interval that the authors use in categorizing meta-analytic evidence. The described problem is important because investigators almost always include more than one nutritional exposure in disease models. This change in nature of the bias in the estimated effect in models with two or more dietary exposures arises from a phenomenon known as residual confounding caused by measurement error. When there are several nutritional intake covariates that are mismeasured, each will adopt a part of the effect of the others, and the fractions of the effects that are adopted will depend on the relative sizes of the errors and the correlations among them. A meta-analysis of studies with similar structures of measurement error may increase the detrimental effects of residual confounding. In the literature, several investigators have cautioned against overinterpreting apparently highly precise results reported from meta-analyses of observational studies. For example, Egger et al. (6) warn that residual confounding can distort findings from observational studies and of the consequent “danger that meta-analyses of observational data produce very precise but equally spurious results.” They conclude that “the statistical combination of data should not therefore be a prominent component of reviews of observational studies.” This concern was expressed for all observational studies where residual confounding may be caused by unknown or unobserved covariates. In nutritional epidemiology, the possibility of residual confounding becomes especially acute due to substantial measurement error (7). It is therefore critical that the present authors carefully discuss effects of dietary measurement error and how it may change the interpretation of their evaluation.

Reply: We thank the reviewer for this detailed comment! We have now included a new paragraph in the Discussion (lines: 391-404) to better describe the sources of error in nutritional epidemiology studies, and a separate paragraph (lines: 405-418) for suggestions to reduce measurement issues in assessing dietary intake. We have rephrased our conclusion statement to better acknowledge the latter issues, as: “Taking into account the inclusion of only observational studies and the limitations of the dietary assessment methods, we found strong or highly suggestive evidence...”

4) Following up on my previous comment, the structure of dietary measurement error depends on both the dietary assessment instrument and the study population to which this instrument is applied. It is therefore important that the authors provide information on the dietary assessment instruments involved in their meta-analyzed studies and the corresponding populations. It would also be of interest if the authors could discuss some suggested ideas of how to mitigate the detrimental effects of dietary measurement error, for example, provided in (7 - 8).

We have now included a new paragraph in the Discussion (lines: 391-404) to better describe the sources of error in nutritional epidemiology studies, and a separate paragraph (lines: 405-418) for suggestions to reduce measurement issues in assessing dietary intake after taking into consideration the reviewer comments and

suggested references. Observational studies in nutritional epidemiology have largely relied on food frequency questionnaires (FFQs) to measure the consumption of the different dietary factors (most studies are in European-decent populations) and a small percentage of studies have included other methods such as 24-hour recalls usually in combination with an FFQ.

REVIEWER COMMENTS

Reviewer #1 (Remarks to the Author):

Dear authors,

thank you for your extensive work on this manuscript and the detailed answers to the comments. The revision improved the manuscript substantially.

Nevertheless, my major concern remains, as few new insights were generated, even though I appreciate the work and the approach to determine, if further studies might add to the field and the discussion of mechanisms of action of nutrients and how to move the field forward.

I would recommend to publish it in an alternative journal due to the scope and the content of this manuscript, which is in my opinion different to the primary scope and articles of this specific journal.

Reviewer #2 (Remarks to the Author):

The Authors are to be commended for their thorough responsiveness to the extensive comments, queries, and suggestions. I found the substantial additions to the text throughout - but especially in the Discussion to address dietary methodological issues - very valuable to the overall manuscript. This includes the improvements to Methods (including Supplemental description of the search methods) and Results, as well as the new tables and conversion of data tables to Excel (very appropriately designed to improve usability for researchers). Overall, the revised manuscript is stronger and more accessible, and its timely publication will be well-received.

Reviewer #3 (Remarks to the Author):

The authors have thoughtfully addressed by questions and concerns. I have no further comments.

Reviewer #5 (Remarks to the Author):

I have been asked to evaluate the methodology of this umbrella review and authors' responses to reviewers' comments. I am impressed by the enormous works undertaken by the authors, but I am particularly intrigued by the calculations of additional studies required for each dietary exposures.

Calculating required sample sizes and statistical power is essential for randomised controlled trials, but for a meta-analysis, those post hoc calculations are much less useful and prone to misinterpretation.

Several issues need to give further clarifications:

1. How is the additional number of studies obtained? Is it based the current point estimate and heterogeneity? What is the sample size/case number in each additional trial?

2. Is it likely that some exposures do not increase the risk of diseases or mortality? In this scenario, the number of studies would be that providing convincing evidence of a null relationship.

REPLY TO REVIEWER COMMENTS

Reviewer #1:

Dear authors,

thank you for your extensive work on this manuscript and the detailed answers to the comments. The revision improved the manuscript substantially.

Nevertheless, my major concern remains, as few new insights were generated, even though I appreciate the work and the approach to determine, if further studies might add to the field and the discussion of mechanisms of action of nutrients and how to move the field forward.

I would recommend to publish it in an alternative journal due to the scope and the content of this manuscript, which is in my opinion different to the primary scope and articles of this specific journal.

We thank the reviewer for assessing our manuscript. Although our research topic has been previously addressed and some associations, such as alcohol intake and breast [1] and colorectal [2] cancers, are established in the literature, we consider our work as highly informative regarding the evidence-synthesis design and methods followed to evaluate the scientific evidence. We believe that our work not only can guide future research in the fields of nutritional epidemiology and cancer prevention but it is structured in a way that can be of interest and inform a wider audience. Furthermore, our work aimed (but was not limited to) to providing insights on the status of current and future research, by identifying the diet-cancer associations in which future research seems futile, unless major improvements in the involved methodologies can take place (see p. 9-10, lines: 383-408, p.10, lines: 424-426). This can be especially interesting for a reader who is not well-versed with the limitations of this field of research.

Reviewer #2:

The Authors are to be commended for their thorough responsiveness to the extensive comments, queries, and suggestions. I found the substantial additions to the text throughout - but especially in the Discussion to address dietary methodological issues - very valuable to the overall manuscript. This includes the improvements to Methods (including Supplemental description of the search methods) and Results, as well as the new tables and conversion of data tables to Excel (very appropriately designed to improve usability for researchers). Overall, the revised manuscript is stronger and more accessible, and its timely publication will be well-received.

We thank the reviewer for the comments.

Reviewer #3:

The authors have thoughtfully addressed by questions and concerns. I have no further

comments.

We thank the reviewer for the comment.

Reviewer #5:

I have been asked to evaluate the methodology of this umbrella review and authors' responses to reviewers' comments. I am impressed by the enormous works undertaken by the authors, but I am particularly intrigued by the calculations of additional studies required for each dietary exposures.

Calculating required sample sizes and statistical power is essential for randomised controlled trials, but for a meta-analysis, those post hoc calculations are much less useful and prone to misinterpretation.

We agree with the reviewer that the potential of misinterpreting the results of the conditional power estimation exists. Yet, we need to stress that the aim of such calculations was not the evaluation of the post-hoc power of the associations. Rather, conditional power analysis in a meta-analysis under this setting should be evaluated under the prism of informing and driving future research [3-5], which we have articulated in several sections of the manuscript (i.e. lines: 56, 80-82, 191-217, 432-435, 533-535). Other reasons for power analysis do exist in meta-analytic research. For example, the evaluation of power and information size in sequential meta-analysis settings, where the scope of power analysis is to account for type 1 error inflation [6-7], or power estimation during the planning stage of a systematic review [8]. Misinterpretation of the results may rise when failing to identify the main focus of each power analysis.

Several issues need to give further clarifications:

1. How is the additional number of studies obtained? Is it based the current point estimate and heterogeneity? What is the sample size/case number in each additional trial?

For conditional power calculations, which are applied in the nominally non-statistically significant observed meta-analyses, the number of new studies is a function of the point estimate and the heterogeneity of the observed meta-analysis, as the reviewer correctly mentions. Regarding the point estimate, as it is described in the methods section (p.12, lines: 535-540), we considered two plausible scenarios: the estimate of the future studies is equal to: a) the summary estimate of the observed meta-analysis and b) the estimate of the largest study in the observed meta-analysis. The findings were similar in these two scenarios. The second component of the conditional power calculation is that the inclusion of the new studies would not alter the heterogeneity of the observed meta-analysis (p.12, lines: 539-540). More details on the methodology and the computations can be found in the original publication [3]. The code for implementing our analysis has been provided to the journal.

Similarly, the fail-safe number (p.12, lines: 541-544), which is applied in the nominally statistically significant observed meta-analyses, represents the number of new studies of mean null association that would need to be added to the observed meta-analysis to produce a t-score at a desired significance level at (e.g., 0.05). This index depends on the number of studies already included in a meta-analysis as well as the critical value t . More details on the relevant computations are available in the original publication by MS Rosenberg [9].

Regarding reviewer's question on sample size, the conditional power methodology was based on information size expressed by the (absolute) weight of the included studies rather than on the study sample size since it assumed that each future study was of average weight as those included in the observed meta-analysis (p.12, lines: 532-544). The same scenario applies to the fail-safe number calculations. Therefore, no sample size/case numbers are calculated as part of the above analyses. Although we acknowledge the potential importance of providing an estimate of the average information size, we refrained from doing so because it would complicate the interpretation of our results. The inputs used for the estimations were data-driven, which seems more plausible compared to making subjective hypotheses regarding the magnitude of future studies' effect size and information size. We have added some clarifications in the Methods (p.12, lines: 532-544) to better reflect the above points.

2. Is it likely that some exposures do not increase the risk of diseases or mortality? In this scenario, the number of studies would be that providing convincing evidence of a null relationship.

In our umbrella review, we verified numerous null associations between dietary factors and risk of cancer in various anatomical sites, and we investigated the number of future studies (p.12, lines: 532-544) required to achieve a conditional power of at least 80% to detect a nominally statistically significant effect size equal to the observed meta-analytic summary estimate assuming that the heterogeneity of the updated meta-analysis did not change. However, an association may truly exist, but the corresponding meta-analysis may fail to capture it for various reasons, such as large between-study heterogeneity, bias, or low number of relevant studies and cancer cases. For the estimation of conditional power, we assumed that each future study provided information equal to the average weight of the included studies. In the case, for example, where the average information size was very low, a large number of future studies would be required in order for an updated random effects meta-analysis to reach adequate conditional power, no matter how homogeneous the included studies were. Although arbitrary, the assumption for the information size of the new studies is realistic as it is based on evidence from the current state of the literature. However, this may not be the case if larger studies or better methods to capture the dietary exposures of interest become available, where in this case a smaller number of studies with larger information size or of better quality would be required.

References

1. World Cancer Research Fund / American Institute for Cancer Research. Breast cancer. How diet, nutrition and physical activity affect breast cancer risk [Available from: <https://www.wcrf.org/dietandcancer/breast-cancer>]
2. World Cancer Research Fund / American Institute for Cancer Research. Colorectal cancer. How diet, nutrition and physical activity affect colorectal cancer risk [Available from: <https://www.wcrf.org/dietandcancer/colorectal-cancer>]
3. Roloff V, Higgins JP, Sutton AJ. Planning future studies based on the conditional power of a meta-analysis. *Stat Med*. 2013 Jan 15;32(1):11-24. doi: 10.1002/sim.5524. Epub 2012 Jul 11. PMID: 22786670; PMCID: PMC3562483.
4. Nikolakopoulou A, Mavridis D, Salanti G. Using conditional power of network meta-analysis (NMA) to inform the design of future clinical trials. *Biom J*. 2014 Nov;56(6):973-90. doi: 10.1002/bimj.201300216. Epub 2014 Sep 16. PMID: 25225031.
5. Nikolakopoulou A, Mavridis D, Salanti G. Planning future studies based on the precision of network meta-analysis results. *Stat Med*. 2016 Mar 30;35(7):978-1000. doi: 10.1002/sim.6608. Epub 2015 Aug 6. PMID: 26250759.
6. Higgins JP, Whitehead A, Simmonds M. Sequential methods for random-effects meta-analysis. *Stat Med*. 2011 Apr 30;30(9):903-21. doi: 10.1002/sim.4088. Epub 2010 Dec 28. PMID: 21472757; PMCID: PMC3107948.
7. Wetterslev J, Thorlund K, Brok J, Gluud C. Trial sequential analysis may establish when firm evidence is reached in cumulative meta-analysis. *J Clin Epidemiol*. 2008 Jan;61(1):64-75. doi: 10.1016/j.jclinepi.2007.03.013. Epub 2007 Aug 23. PMID: 18083463.
8. Jackson D, Turner R. Power analysis for random-effects meta-analysis. *Res Synth Methods*. 2017 Sep;8(3):290-302. doi: 10.1002/jrsm.1240. Epub 2017 Apr 4. PMID: 28378395; PMCID: PMC5590730.
9. Rosenberg MS. The file-drawer problem revisited: a general weighted method for calculating fail-safe numbers in meta-analysis. *Evolution*. 2005 Feb;59(2):464-8. PMID: 15807430.

REVIEWER COMMENTS

Reviewer #5 (Remarks to the Author):

The authors have addressed my comments carefully; I have no further comments.

Response to reviewer comments

Reviewer comments

Reviewer #5:

The authors have addressed my comments carefully; I have no further comments.

Reply: Thank you